# Surgical outcomes after primary Baerveldt glaucoma implant surgery with vitrectomy for neovascular glaucoma

Koichi Nishitsuka[ID]°, Akira Sugano°*, Takayuki Matsushita, Katsuhiro Nishi[ID], Hidetoshi Yamashita

Department of Ophthalmology and Visual Sciences, Yamagata University Faculty of Medicine, Yamagata, Japan

° These authors contributed equally to this work.
* mlc12186@nifty.com

**Data Availability Statement:** All relevant data are within the paper and its Supporting information files.

## Abstract

This study aimed to evaluate the 3-year long-term outcomes of primary Baerveldt glaucoma implant (BGI) surgery for neovascular glaucoma (NVG). We retrospectively evaluated 27 consecutive patients with NVG between November 2013 and November 2017. All the patients were treated with panretinal photocoagulation and pars plana vitrectomy before BGI surgery without anti-vascular endothelial growth factor treatment. The surgical success of the BGI was defined as an IOP of <22 mmHg and <5 mmHg with or without antiglaucoma medication. The outcomes were assessed on the basis of intraocular pressure (IOP), visual acuity, postoperative complications, and cumulative success rate. Except for 2 mortality cases, 25 eyes (92.6%) were followed up for 3 years. The mean IOPs (mmHg)/numbers of glaucoma medications ± standard error of the mean before and 12 and 36 months after BGI surgery were 41.6/4.6 ± 1.9/0.2, 14.8/2.2 ± 0.8/0.4 and 16.9/2.6 ± 1.1/0.3, respectively. In all of the follow-up time points, the postoperative mean IOP and number of glaucoma medications were statistically significantly lower than the preoperative values (analysis of variance, P < 0.001). At 3 years after surgery, the rates of visual acuity improvement (logMAR ≤ −0.3), invariance (−0.3 < logMAR < 0.3), and worsening (logMAR ≥ 0.3) were 56.0% (14/25 eyes), 24.0% (6/25 eyes), and 20.0% (5/25 eyes), respectively. The most common postoperative complications were hyphema (4 eyes, 14.8%) and vitreous hemorrhage (5 eyes, 18.5%), and serious complications such as expulsive hemorrhage, endophthalmitis, and tube/plate exposure did not occur. The cumulative probabilities of surgical success after the operation were 100% at 1 year, 85.2% at 2 years, and 77.4% at 3 years. In conclusion, combined non-valved pars plana tube placement in conjunction with vitrectomy was successful at lowering IOP with relatively low complication rates.

**Funding:** This work was supported by JSPS KAKENHI Grant Numbers JP25462704, JP20K18373.

# Introduction

Neovascular glaucoma (NVG) is a serious disease that is generally resistant to surgical and drug treatments and ultimately leads to blindness [1]. As the high intraocular pressure (IOP) associated with NVG not only destroys the remaining visual function but also causes eye pain, reducing the IOP in addition to the treatment of the causative disease is important. The choice of surgical treatment for glaucoma has expanded in recent years, and tube shunt surgery, which is one of the treatments for intractable glaucoma, has the similar effect of lowering IOP as trabeculectomy in the Tube versus Trabeculectomy (TVT) Study [2–4].

Tube shunt surgery was generally positioned as a last-resort treatment when classic trabeculectomy treatment was unsuccessful, but in recent years, tube shunt surgery has been selected as an alternative to trabeculectomy [5–9]. The Primary Tube versus Trabeculectomy (PTVT) Study is a randomized clinical trial comparing trabeculectomy and initial tube shunt surgery in 242 glaucoma patients who had poor IOP control and had not undergone surgical treatment [10]. In addition, a meta-analysis showed that tube shunt surgery is effective for the treatment of NVG [11]. Although primary tube shunts may be effective against NVG, which is an intractable disease, so far, the primary tube has been used for NVG. In this study, we will report the 3-year treatment results of using the primary tube as the initial glaucoma treatment for NVG and performing Baerveldt glaucoma implant (BGI) surgery with vitrectomy.

# Materials and methods

The institutional review committee of the Yamagata University Faculty of Medicine approved the study protocol (approval No. S-61). The study was performed in accordance with the tenets of the Declaration of Helsinki. Written informed consent about the NVG was obtained from all patients. In this study, we investigated 27 eyes of 27 consecutive patients with NVG who underwent BGI surgery as the primary glaucoma surgery at Yamagata University Hospital between November 2013 and November 2017. We retrospectively reviewed the medical records of the patients. Patients with a history of glaucoma surgery and no-light-perception vision at the onset of NVG were excluded from the study.

In all the cases, panretinal photocoagulation was first performed as much as possible before surgery. After that, vitreous surgery was performed to add retinal photocoagulation in the outermost part and, if necessary, to treat the underlying diseases such as with a proliferative membrane and bleeding control. After that, BGI surgery (BG 102–350; Johnson & Johnson Surgical Vision, Inc., Santa Ana, CA, USA) was performed. In the BGI surgery, a 180˚ conjunctiva was incised to secure the extraocular muscle and BGI insertion space (Fig 1A). When creating the three 25-gauge ports, 1 port was matched to the BGI Hoffman elbow insertion site (Fig 1B). Peripheral vitreous shaving was performed sufficiently (Fig 1C) to avoid vitreoretinal complications. Before fixing the Baerveldt implant to the eyeball, tube ligation with 8–0 Vicryl was performed (Fig 1D). After removing the port, the incision was widened with a 20-gauge V lance (Fig 1E), and then a Hoffman elbow was inserted. The BGI was fixed to the eyeball with 8–0 nylon (Fig 1F), and a Sherwood slit was made with a needle (Fig 1G). The Hoffman elbow was patched with 8–0 nylon on a preserved sclera (Fig 1H). After confirming that no complications occurred, the remaining 2 ports were closed, and conjunctival sutures were performed.

During the study period, anti-vascular endothelial growth factor (VEGF) treatment had not been approved in the Japanese insurance system, so none of the patients receive anti-VEGF therapy as adjuvant therapy. One vitreoretinal surgeon (K.N.) and one glaucoma surgeon (A.S.) performed all the surgical procedures.

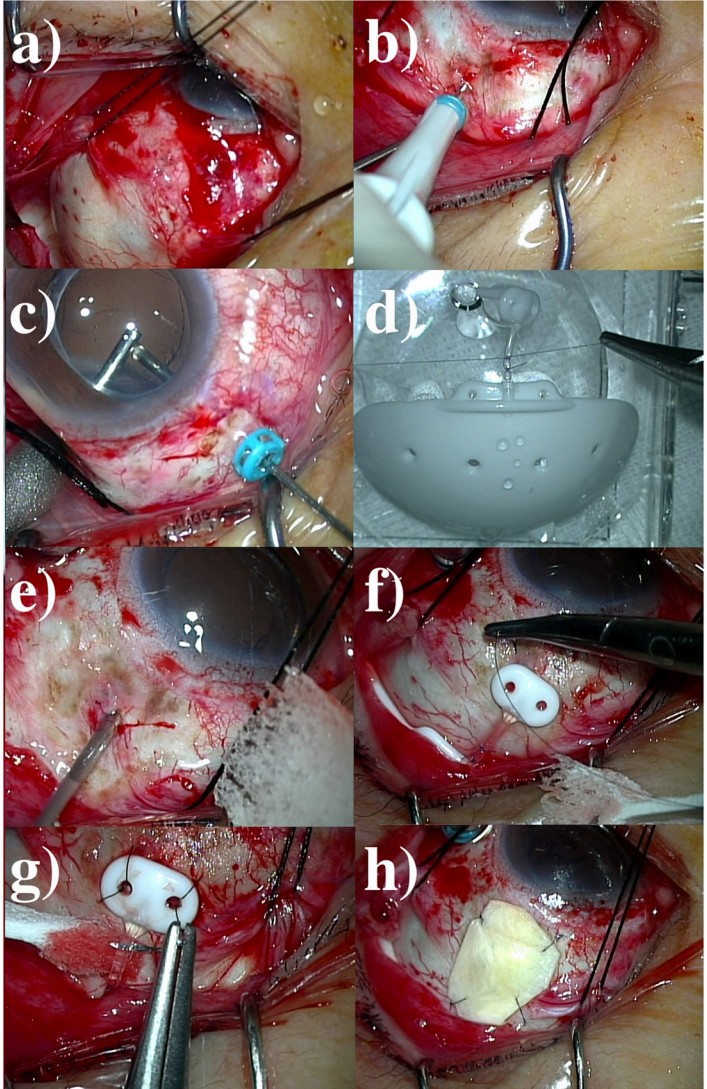

**Fig 1. Intraoperative finding of Baerveldt glaucoma implant surgery in this study.** a) Deployment of the surgical field with a 180˚ conjunctival incision. b) Trocar insertion in the Baerveldt implant fixation site. c) Peripheral vitreous shaving. d) Tube ligation with 8–0 Vicryl. e) A widening incision with a 20-gauge V lance. f) Fixation of the Baerveldt implant to the eyeball. g) Making a Sherwood slit. h) Patching the Hoffman elbow with a preserved sclera.

## Data collection

The patient demographic data collected were as follows: age, sex, laterality of the eyes, IOP, best-corrected visual acuity (BCVA), number of glaucoma medications, underlying diagnosis of NVG, lens status, and history of ocular surgery. Moreover, informations on intraoperative complications, postoperative complications, reoperation, and reasons for treatment failure were collected. We measured the IOP using a Goldmann applanation tonometer. We measured the IOP and number of medications preoperatively and at 1, 3, 6, 12, 18, 24, 30, and 36 months after the primary glaucoma tube shunt surgery. The total numbers of medications were as follows: single anti-glaucoma eye drops (1 medication; prostaglandin analogs, beta blockers, carbonic anhydrase inhibitors, alpha agonists, Rho kinase inhibitors), combined anti-glaucoma eye drops (2 medications; beta blocker & carbonic anhydrase inhibitor and

prostaglandin & beta blocker), and acetazolamide oral medicine (2 medications). The surgical success of the BGI was defined as an IOP of <22 mmHg and <5 mmHg with or without anti-glaucoma medication.

For the BCVA measurement, we used a Japanese decimal visual acuity chart placed 5 m away from the patient. The decimal visual acuity was converted into Snellen visual acuity and logarithmic minimum angle of resolution (logMAR) to examine the visual acuity change. We compared the decimal visual acuity distribution and the amount of vision change between the preoperative, 1-year postoperative, and 3-year postoperative periods. An increase of ≥0.3 log-MAR unit, a change of <0.3 logMAR unit, and a decrease of ≥0.3 logMAR unit in comparison with the preoperative value were defined as "worsenig," "invariant," and "improvement," respectively.

## Statistical analyses

The analysis of variance (ANOVA) and Fisher exact test were used for the statistical analyses. In all the analyses, P values < 0.01 were considered statistically significant. Surgical failure was determined using the Kaplan-Meier life-table analysis. The time to failure was defined as the time from treatment to reoperation for glaucoma, loss-of-light-perception vision, or the first of 2 consecutive study visits after 3 months when the patient showed persistent hypotony (IOP < 6 mmHg) or hypertension (IOP > 21 mmHg). All statistical data were analyzed using PASW Statistics 18 (SPSS Inc., Chicago, IL, USA).

## Results

The baseline patient characteristics of the patients are shown in Table 1. Two patients accounting for 2 eyes died during the course, so only 25 eyes from 25 patients could be followed up for >3 years after a primary glaucoma surgery. The breakdown was 18 men and 9 women, with a mean age of 56.4 ± 14.3 years. The breakdown of the causative diseases of NVG was proliferative diabetic retinopathy (PDR) with 16 eyes, ocular ischemic syndrome (OIS) with 8 eyes, and retinal vein occlusion (RVO) with 3 eyes. The preoperative mean IOP, mean BCVA, and

**Table 1. Patients' demographics.**

| Number of patients/eyes | | 27/27 |
| --- | --- | --- |
| Age (years), mean ± SD (range) | | 56.4 ± 14.3 (25–84) |
| Sex (female/male) | | 9/18 |
| Laterality (right/left) | | 16/11 |
| Underlying diagnosis of NVG, eyes | PDR | 16 (59.3%) |
| | OIS | 8 (29.6%) |
| | RVO | 3 (11.1%) |
| Preoperative IOP, mmHg, mean ± SD (range) | | 41.6 ± 9.9 (24–60) |
| Preoperative BCVA, LogMAR, mean ± SD (range) | | 1.4 ± 0.9 (0–3) |
| Number of glaucoma medications, mean ± SD (range) | | 4.6 ± 1.0 (3–6) |
| Lens status (phakia/pseudophakia) | | 13/14 |
| History of ocular surgery | Cataract surgery | 14 (51.9%) |
| | Vitrectomy | 7 (25.9%) |
| | PRP | 17 (63.0%) |

SD: standard deviation; NVG: neovascular glaucoma; PDR: proliferative diabetic retinopathy; OIS: ocular ischemic syndrome; RVO: retinal vein occlusion; BCVA: best-corrected visual acuity; logMAR: logarithmic minimum angle of resolution; PRP: panretinal photocoagulation

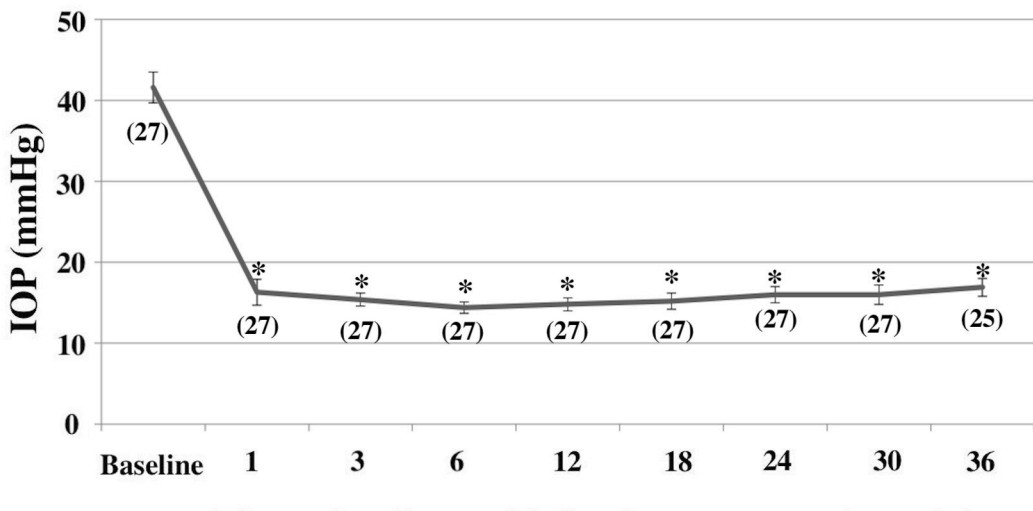

**Fig 2. Postoperative intraocular pressure course after Baerveldt implant surgery.** The mean intraocular pressures (IOP) ± standard error of the mean (± SEM) during the preoperative and 1-, 3-, 6-, 12-, 18-, 24-, 30-, and 36-month postoperative periods after Baerveldt implant surgery are shown. *P < 0.001, compared with the preoperative mean IOP based on the analysis of variance with Bonferroni correction. The number of eyes at each time point is shown in parentheses.

mean number of medications ± standard deviation (SD) were 41.6 ± 9.9 mmHg, 1.4 ± 0.9, and 4.6 ± 1.0, respectively.

The baseline and follow-up IOPs are shown in Fig 2. The mean IOP ± standard error of the mean (SEM) during the preoperative and 12-, and 36-month postoperative periods after Baerveldt implant surgery were 41.6 ± 1.9, 14.8 ± 0.8, and 16.9 ± 1.1 mmHg, respectively. In all of the follow-up time points, the postoperative mean IOPs were statistically significantly lower than the preoperative mean IOPs (ANOVA, P < 0.001).

The numbers of glaucoma medications at baseline and follow-up are shown in Fig 3. The mean numbers of glaucoma medications ± SEM during the preoperative and 12-, and 36-month postoperative periods were 4.6 ± 0.2, 2.2 ± 0.4, and 2.6 ± 0.3, respectively. At all of the follow-up time points, the postoperative mean number of glaucoma medications was statistically significantly lower than the preoperative mean number of glaucoma medications (ANOVA, P < 0.001). In addition, in all the patients who were taking acetazolamide oral medication preoperatively, the therapy was discontinued after surgery.

Table 2 shows the distributions of the visual acuity improvement, invariance, and worsening at 1 and 3 years postoperatively. At 1 year after surgery, the rates of visual acuity improvement (logMAR ≤ −0.3), invariance (−0.3 < logMAR < 0.3), and worsening (logMAR ≥ 0.3) were 40.7% (11/27 eyes), 40.7% (11/27 eyes), and 18.5% (5/27 eyes). At 3 years after surgery, the rates were 56.0% (14/25 eyes), 24.0% (6/25 eyes), and 20.0% (5/25 eyes) were obtained, respectively.

The incidence rates of intraoperative complications, early postoperative complications (postoperative < 6 months), long-term complications (postoperative ≥ 6 months), and reoperation are described in Table 3. Intraoperative complications occurred during vitreous surgery performed a few days after prone air replacement in 1 patient who was undergoing retinal dialysis; BGI surgery was performed after the fundus lesions had resolved. No serious complications such as expulsive hemorrhage occurred. The most common postoperative complications in the early period were hyphema (4 eyes, 14.8%) and vitreous hemorrhage (5 eyes,

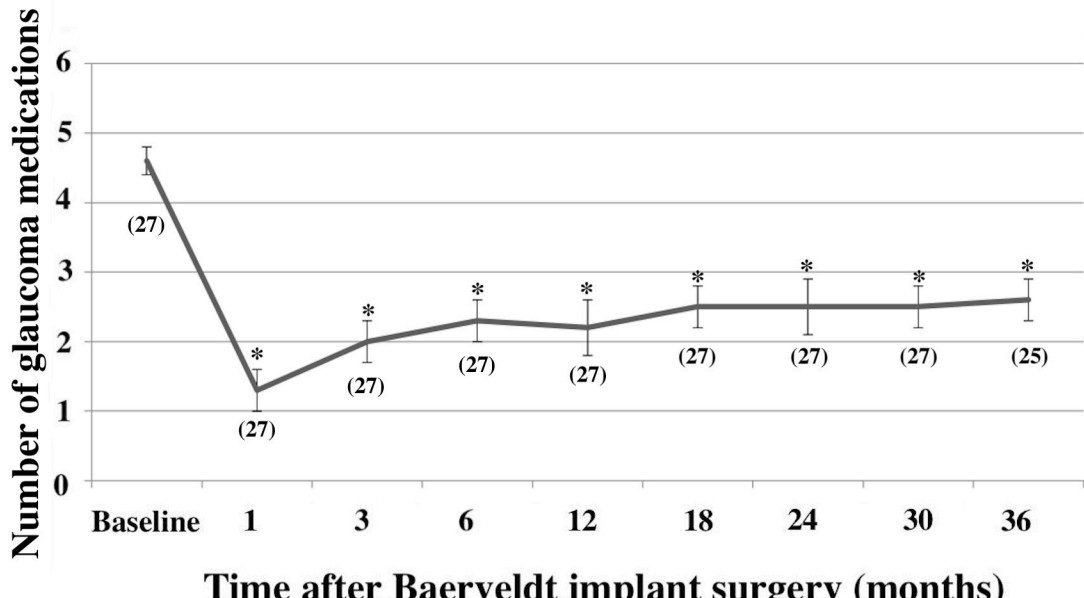

**Fig 3. Postoperative transition of the number of glaucoma medications.** The mean numbers of glaucoma medications ± standard error of the means (SEM) during the preoperative and 1-, 3-, 6-, 12-, 18-, 24-, 30-, and 36-month after Baerveldt implant surgery are shown. *P < 0.001, compared with the preoperative mean number of glaucoma medications based on the analysis of variance with Bonferroni correction. The number of eyes at each time point is shown in parentheses.

18.5%). The postoperative complications in the late period included noninfectious iritis in 4 eyes (14.8%), which all improved with steroid eye drops, and vitreous hemorrhage in 2 eyes (7.4%). Vitreous reoperation for vitreous hemorrhage was performed in 2 eyes. One case became a traction retinal detachment with low IOP after multiple vitreous surgeries for vitreous hemorrhage.

Fig 4 shows the Kaplan-Meier survival curves. The cumulative probability rate of surgical success after operation was 100% at 1 year, 85.2% at 2 years, and 77.4% at 3 years. The cause of surgical failure was ocular hypertension in 4 eyes (66.6%), low intraocular pressure in 1 eye (16.7%), and light-perception consumption in 1 eye (16.7%). Surgical failure occurred within 1 year after surgery in no eyes, at 2 years after surgery in 4 eyes, at 3 years after surgery in 2 eyes, and unknown owing to death.

## Discussion

In this study, we report the treatment results in consecutive cases of NVG in which glaucoma tube shunt surgery (BGI) was chosen as the first-line glaucoma surgery. The mean

**Table 2. Distributions of visual acuity improvement, invariance, and deterioration at 1 and 3 years after surgery.**

|  | Improvement* | Invariant** | Worsen*** | P value |
|---|---|---|---|---|
| 1 year after surgery (n = 27) | 11 (40.7%) | 11 (40.7%) | 5 (18.5%) | 0.416 |
| 3 years after surgery (n = 25) | 14 (56.0%) | 6 (24.0%) | 5 (20.0%) |  |

*The postoperative BCVA improved by ≥0.3 LogMAR unit as compared with the preoperative BCVA.

**The postoperative BCVA changed within 0.3 LogMAR unit as compared with the preoperative BCVA.

***The postoperative BCVA worsened by ≥0.3 LogMAR unit as compared with the preoperative BCVA.

BCVA: best-corrected visual acuity; LogMAR: logarithmic minimum angle of resolution

**Table 3. Intraoperative complications, postoperative complications, and reoperation.**

|  |  | No. of eyes (%) |
| --- | --- | --- |
| Intraoperative complications | Retinal dialysis | 1 (3.1%) |
|  | Expulsive hemorrhage | 0 (0%) |
| Early postoperative complications* | Wound leak | 0 (0%) |
|  | Hypotony | 0 (0%) |
|  | Hyphema | 4 (14.8%) |
|  | Endophthalmitis | 0 (0%) |
|  | Rhegmatogenous retinal detachment | 0 (0%) |
|  | Vitreous hemorrhage | 5 (18.5%) |
| Late postoperative complications** | Diplopia | 0 (0%) |
|  | Hypotony | 1 (3.7%) |
|  | Iritis | 4 (14.8%) |
|  | Endophthalmitis | 0 (0%) |
|  | Vitreous hemorrhage | 2 (7.4%) |
|  | Traction retinal detachment | 1(3.7%) |
|  | Plate and tube exposure | 0 (0%) |
|  | Bullous keratopathy | 0 (0%) |
| Reoperation | Vitrectomy | 2 (7.4%) |
|  | Glaucoma surgery | 0 (0%) |

*Complications < 6 month after surgery

**Complications ≥ 6 months after surgery

preoperative IOP was 41.6 ± 1.9 mmHg, and the mean 1-month postoperative IOP was 16.3 ± 1.6 mmHg, showing a decrease in IOP in all the cases. Three years after the operation, 75.0% of the cases met the success criteria, and 56.0% of the cases showed improvement in visual acuity. Several studies that investigated the 3-year surgical outcomes of NVG treatment, including this study, are shown in Table 4. The success rates of BGI [12], Ahmed valve implant [13, 14], and trabeculectomy (TLE) [15, 16] at 3 years after surgery were 75.0%–78.1%, 20.6%–62.5%, and 82.8%–83.3%, respectively. The results were considered to vary depending on the treatment method and presence or absence of concomitant use of anti-VEGF treatment procedure [17]. In glaucoma tube shunt surgery, treatment outcomes may also be affected, especially in patients with or without a history of glaucoma surgery, including TLE. As far as we know, no reports have described a long-term course of 3 years with a high follow-up rate in consecutive cases of NVG in which BGI was selected as the primary tube shunt surgery. This study demonstrates the effectiveness of primary tube shunt surgery in NVG eyes.

In this study, we chose a vitreous cavity insertion-type BGI for NVG and adopted a strategy in which the retinal vitreous and glaucoma surgeons cooperated to perform the surgery. Primary tube surgery is easier to insert glaucoma implants than secondary tube surgery after TLE. The reason is that the conjunctiva is preserved in primary tube surgery. This point is considered to be an advantage of primary tube surgery. In addition, by combining with vitrectomy, retinal photocoagulation, including the peripheral retina for ischemic retina, and PDR treatment, including removal of bleeding and proliferative membranes, could be performed sufficiently.

The TVT study [2] reported the following complication rates: hyphema, 2%; vitreous hemorrhage, 1%; endophthalmitis, 1%; tube erosion, 5%; and retinal detachment, 1%. Meanwhile, the PTVT study [10] reported 6% and 1% incidence rates of hyphema and plate exposure, respectively. In addition, Anton et al [18] reported the following incidence rates: hyphema,

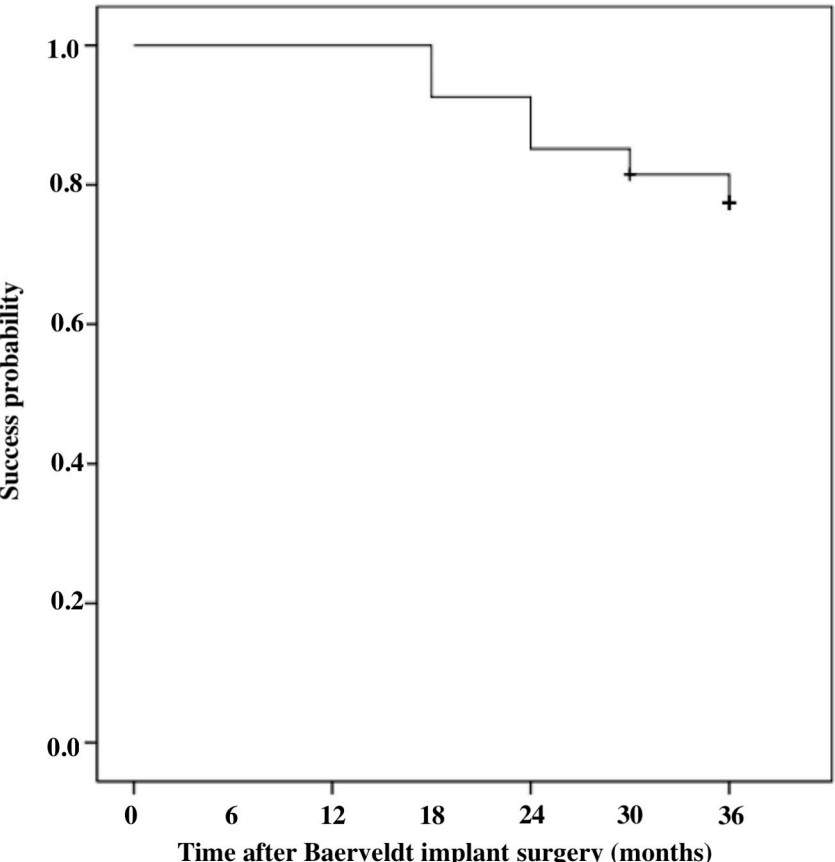

**Fig 4. Kaplan-Meier survival analysis of surgical success.**

21%; vitreous hemorrhage, 16%; endophthalmitis, 3.4%; and retinal detachment, 4.5%, in BGI surgery with vitrectomy. In tube shunt surgery for NVG, the incidence rates of vitreous hemorrhage and hyphema seem high. In this study, the incidence rates of vitreous hemorrhage and hyphema were 18.8% and 18.8%, respectively (Table 3). On the other hand, no complications

**Table 4. Surgical success of 3-year follow-up in neovascular glaucoma.**

|  | N | Procedure | Anti-VEGF | 3-year follow-up (%) | 3-year cumulative probability (%) |
|---|---|---|---|---|---|
| Present study | 27 | BGI | - | 92.6 | 77.4[a] |
| Tojo et al. [12] | 35 | BGI | + | 25.7 | 78.1[a] |
| Park et al. [13] | 42 | AI | + | 26.2 | 62.5[a] |
| Netland et al. [14] | 38 | AI | - | 18.4 | 20.6[a] |
| Kobayasahi et al. [15] | 12 | TLE | + | 100 | 83.3[b] |
| Higashide et al. [16] | 61 | TLE | + | 57.4 | 82.8[a] |
| Noor et al. [17] | 9 | Glaucoma Implant* | + | unknown | 53.6[a] |
| Noor et al. [17] | 30 | Glaucoma Implant* | - | unknown | 31.6[a] |

BGI: Baerveldt glaucoma implant, TLE: Trabeculectomy, AI: Ahmed valve implant.

*Glaucoma implant, including the Baerveldt glaucoma, Ahmed valve, and Keiki Mehta implants

[a] Surgical success was defined as 22 mmHg> IOP >5 mmHg.

[b] Surgical success was defined as 22 mmHg> IOP.

of endophthalmitis, retinal detachment, and plate/tube exposure were found in this study (Table 3). We believe that sharing treatment strategies in the fields of glaucoma and vitreoretina from the initial surgery is the factor of success. The absence of plate/tube exposure as a complication may have resulted from the adequate surgical field and use of preserved sclera for patching in all the cases.

The effectiveness of the adjuvant use of anti-VEGF has been reported in TLE for NVG [15, 16]. Zhou et al. reported that anti-VEGF is effective for the treatment success and prevention of bleeding complications in a meta-analysis [19]. On the other hand, Noor et al. reported no significant differences in the postoperative IOP, number of glaucoma medications, and success rate with or without the use of anti-VEGF in tube shunt surgery for NVG [17]. Anti-VEGF was not used in this study because it had not been approved during the study period. On the basis of previous reports, the use of anti-VEGF may reduce bleeding complications in BGI surgery, but further studies are needed on the final therapeutic effect.

The strategy of BGI surgery may also be important for long-term results. In this study, primary BGI surgery was chosen as the first glaucoma surgery for NVG, and the tubes were inserted in the pars plana in all the patients. Iwasaki et al. reported that the corneal endothelium decreased by 13.1% at 1 year after the operation in the case of BGI inserted in the anterior chamber [20]. On the other hand, no significant decrease in corneal endothelium was observed in the group in which BGI was inserted in the pars plana [20]. Koo et al. also reported that tubes closer to the cornea seemed to lead to increased loss of adjacent endothelial cells [21]. Zhang et al. compared more than 100 Ahmed valve implant cases of sulcus and anterior chamber tube placement and showed that tube location in anterior chamber were associated with faster central endothelial cell density loss [22]. As our treatment strategy, retinal vitreous and glaucoma specialists collaborated to perform vitrectomy and BGI surgery with pars plana insertion in all the cases. These treatments were considered good methods for avoiding the risk of corneal endothelium loss in addition to treating fundus diseases, including ischemic retina.

However, the present study has several limitations, including its retrospective design and lack of a control group. An additional period of follow-up is required to assess long-term prognosis, including tube exposure. Comparison with TLE, the types of glaucoma implants, glaucoma surgery history, and combination with anti-VEGF treatment should be further evaluated. As administration of anti-VEGF to NVG is currently permitted in Japan, a thorough examination of the administration protocol and investigation of its effectiveness in primary glaucoma implant surgery are necessary.

In summary, we evaluate the 3-year long-term outcomes of primary BGI surgery for NVG. Our results suggest that combined non-valved pars plana tube placement in conjunction with vitrectomy is successful at lowering IOP with relatively low complication rates. In the strategy of primary tube surgery combined with vitrectomy, collaboration between glaucoma and retina specialists may be important.

## Supporting information

**S1 Data.**
(XLSX)

## Author Contributions

**Conceptualization:** Koichi Nishitsuka, Akira Sugano.

**Data curation:** Koichi Nishitsuka, Akira Sugano, Takayuki Matsushita, Katsuhiro Nishi.

**Formal analysis:** Koichi Nishitsuka, Akira Sugano, Takayuki Matsushita, Katsuhiro Nishi.

**Funding acquisition:** Koichi Nishitsuka, Katsuhiro Nishi.

**Supervision:** Koichi Nishitsuka, Hidetoshi Yamashita.

**Writing – original draft:** Koichi Nishitsuka, Akira Sugano.

**Writing – review & editing:** Koichi Nishitsuka, Hidetoshi Yamashita.

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
