## [Decision Letter · Decision Letter 0]

17 Mar 2021

PONE-D-21-04967

Surgical outcomes after primary Baerveldt glaucoma implant surgery with vitrectomy for neovascular glaucoma

PLOS ONE

Dear Dr. Nishitsuka,

Thank you for submitting your manuscript to PLOS ONE. After careful consideration, we feel that it has merit but does not fully meet PLOS ONE’s publication criteria as it currently stands. Therefore, we invite you to submit a revised version of the manuscript that addresses the points raised during the review process.

ACADEMIC EDITOR:

The manuscript needs careful English editing for grammar. There are some statistical bias by using both eyes of the same patient. In addition, there are some bias in the choice of papers for comparison with the outcome in the discussion section.

We look forward to receiving your revised manuscript.

Kind regards,

Ahmed Awadein, MD, Ph.D, FRCS

Academic Editor

PLOS ONE

Journal Requirements:

Additional Editor Comments (if provided):

Reviewers' comments:

Reviewer's Responses to Questions

**Comments to the Author**

1. Is the manuscript technically sound, and do the data support the conclusions?

Reviewer #1: Yes

Reviewer #2: Yes

Reviewer #3: Partly

2. Has the statistical analysis been performed appropriately and rigorously? 

Reviewer #1: Yes

Reviewer #2: Yes

Reviewer #3: Yes

3. Have the authors made all data underlying the findings in their manuscript fully available?

Reviewer #1: Yes

Reviewer #2: Yes

Reviewer #3: Yes

4. Is the manuscript presented in an intelligible fashion and written in standard English?

Reviewer #1: Yes

Reviewer #2: Yes

Reviewer #3: No

5. Review Comments to the Author

Reviewer #1: - In line 74 : Please correct "Rectus Femoris"

- The authors did not mention any use of anti-VEGF in the 3 years of follow up after surgery, it is better to mention the name of drugs used , doses and number

Reviewer #2: I read with interest Nishitsuka and colleagues' manuscript in which they retrospectively report on the outcomes of Baerveldt surgery in neovascular glaucoma. The article is interesting and I only have minor suggestions for improvement.

1) Including bilateral cases induces a statistical bias. Considering that both eyes were included for only 5 patients, I would recommend the authors exclude 5 of these eyes, retaining only the first operated eye.

2) For the sake of clarity, intraocular pressure and medications do not need to be provided in the abstract and the full text for every single timepoint: baseline, 12 months and 36 months will be enough. Readers can see other results in the figures.

3) If the authors decide to include the success rate in the abstract, they should also detail their definition of success in the abstract too.

4) The definition of success should be made more obvious in the full text.

5) The following sentence: "all the patients did not receive anti VEGF therapy" suggests that some patients received anti-VEGF. If this is not the case, the authors should clarify this ("none of the patients..." would make a clearer wording). If this is the case, they should report how many patients received anti-VEGF and provide a sub-analysis of their outcomes.

6) Lines 110-113: An increase in LogMAR should actually be considered a worsening of the visual acuity (and conversely).

7) In the discussion, if the authors elect to compare success rates between studies, they should make it clear in the text and the table that every study used in the comparison relied on the same success criteria as the present study. If this is not the case, the authors may still choose to report the success rates of these studies, but should discuss the impact of definition discrepencies. They may also consider reporting more comparable factors such as the percentages of intraocular pressure and medication reduction.

8) Occasional syntax errors may be addressed through careful proofreading.

Reviewer #3: This is was a retrospective, single-center, single surgeon pair case series aiming to describe outcomes in patients undergoing primary non-valved pars plana tube placement with concomitant vitrectomy in patients with neovascular glaucoma. This study is of some clinical interest as these surgeries are not frequently done at the same time and valved glaucoma tube shunts are perhaps more frequently used in patients with NVG.

Abstract: With regards to abstract, it would be helpful to define the success/failure criteria as it is only described later. I think too much space in the abstract is given to a breakdown of the complications, which could better be discussed later.

Introduction: The paper as a whole would benefit from proof-reading by a native English speaker. However, this does not detract from the underlying readability, except in a few instances (particularly at lines 104-106).

For my own clarification, lines 45-48 seem to suggest that the TVT study showed the same outcomes, when I would argue they were similar and there were differences between the tube and trabeculectomy groups.

Materials and Methods: It might be more statistically sound to only use 1 eye from any 1 patient. In this study there are several patients that have both eyes enrolled.

It would of course be up to the editor, but the lengthy description of the surgical technique may be better suited for an addendum as it does not add much to the paper.

Discussion: The success rates of the Baerveldt, Ahmed, and trabeculectomy are pulled from a few limited case series and I am concerned that there is the risk for bias. Success may be measured differently between these studies and the authors run the risk of subconsciously choosing studies that support their conclusion that the Baerveldt is highly successful.

While the authors acknowledge their reasons for not using anti-VEGF pre/post-operatively, the standard of care for NVG now almost always includes anti-VEGF. The conclusions of the study may be outdated because of this.

The study may benefit from more comparison of complication rates with other tube insertion techniques (like anterior chamber placement)

Lines 253-254 are difficult to understand

Lines 254-256: The reasoning for the lack of tube exposure after only 3 years may be of limited utility as tube exposure often happens even further out than 3 years

Only one small study is cited for the superiority of pars plana tubes in protecting corneal endothelium.

Overall, the conclusions drawn by the authors in this study suggest that combined non-valved pars plana tube placement in conjunction with vitrectomy is successful at lowering IOP with relatively low complication rates.

6. PLOS authors have the option to publish the peer review history of their article (what does this mean?). If published, this will include your full peer review and any attached files.

Reviewer #1: **Yes: **Mahmoud Rateb

Reviewer #2: No

Reviewer #3: No

---

## [Author Response · Author response to Decision Letter 0]

20 Mar 2021

Reviewer #1:

 - In line 74 : Please correct "Rectus Femoris"

Response: Thank you for your careful peer review. Accordingly, we have revised the manuscript as follow: 

Page 5, Lines 74-75

In the BGI surgery, a 180°conjunctiva was incised to secure the extraocular muscle and BGI insertion space (Fig. 1A).

- The authors did not mention any use of anti-VEGF in the 3 years of follow up after surgery, it is better to mention the name of drugs used , doses and number

Response: We thank the reviewer for this comment. We have described the glaucoma medications used in this study in Methods as follows. And number of glaucoma medications were shown in Figure 3.

Page 7, Lines 105-109

The total numbers of medications were as follows: single anti-glaucoma eye drops (1 medication；prostaglandin analogs, beta blockers, carbonic anhydrase inhibitors, alpha agonists, Rho kinase inhibitors), combined anti-glaucoma eye drops (2 medications; beta blocker & carbonic anhydrase inhibitor and prostaglandin & beta blocker), and acetazolamide oral medicine (2 medications).

 

Reviewer #2: I read with interest Nishitsuka and colleagues' manuscript in which they retrospectively report on the outcomes of Baerveldt surgery in neovascular glaucoma. The article is interesting and I only have minor suggestions for improvement.

Response: We wish to thank the reviewer for reviewing our study. We have revised manuscript according to REVIEWER 2.

1) Including bilateral cases induces a statistical bias. Considering that both eyes were included for only 5 patients, I would recommend the authors exclude 5 of these eyes, retaining only the first operated eye.

Response: We thank the reviewer for this comment. We received a similar point from Reviewer 3. Accordingly, we have exclude 5 eyes and revised the manuscript including Table 1-4 and Figure 2-3. The results do not change much.

2) For the sake of clarity, intraocular pressure and medications do not need to be provided in the abstract and the full text for every single timepoint: baseline, 12 months and 36 months will be enough. Readers can see other results in the figures.

Response: We thank the reviewer for this comment. Accordingly, we have revised the manuscript as follow: 

Page 2, Limes 27-29 (Abstract)

The mean IOPs (mmHg)/numbers of glaucoma medications ± standard error of the mean before and 12 and 36 months after BGI surgery were 41.6/4.6 ± 1.9/0.2, 14.8/2.2 ± 0.8/0.4 and 16.9/2.6 ± 1.1/0.3, respectively.

Page 10, Lines 145-147 (Results)

The mean IOP ± standard error of the mean (SEM) during the preoperative and 12-, and 36-month postoperative periods after Baerveldt implant surgery were 41.6 ± 1.9, 14.8 ± 0.8, and 16.9 ± 1.1 mmHg, respectively.

Page 11, Lines 158-160 (Results)

The numbers of glaucoma medications at baseline and follow-up are shown in Fig 3. The mean numbers of glaucoma medications ± SEM during the preoperative and 12-, and 36-month postoperative periods were 4.6 ± 0.2, 2.2 ± 0.4, and 2.6 ± 0.3, respectively.

3) If the authors decide to include the success rate in the abstract, they should also detail their definition of success in the abstract too.

Response: We thank the reviewer for this comment. Accordingly, we have added the definition of success in Abstract as follow:

Page 2, Lines 23-24 (Abstract)

The surgical success of the BGI was defined as an IOP of <22 mmHg and <5 mmHg with or without antiglaucoma medication.

4) The definition of success should be made more obvious in the full text.

Response: We thank the reviewer for this comment. Accordingly, we have added the definition of success in Methods as follow:

Page 7, Lines 110-111 (Methods)

The surgical success of the BGI was defined as an IOP of <22 mmHg and <5 mmHg with or without antiglaucoma medication.

5) The following sentence: "all the patients did not receive anti VEGF therapy" suggests that some patients received anti-VEGF. If this is not the case, the authors should clarify this ("none of the patients..." would make a clearer wording). If this is the case, they should report how many patients received anti-VEGF and provide a sub-analysis of their outcomes.

Response: We thank the reviewer for this comment. To clarify, we have revised the manuscript as follow:

Page 6, Lines 93-96

During the study period, anti-vascular endothelial growth factor (VEGF) treatment had not been approved in the Japanese insurance system, so none of the patients receive anti-VEGF therapy as adjuvant therapy.

6) Lines 110-113: An increase in LogMAR should actually be considered a worsening of the visual acuity (and conversely).

Response: We thank the reviewer for this comment. The reviewer's comment is correct. To clarify, we have revised the manuscript as follow:

Page 7, Line115 to Page 8, Line 119

We compared the decimal visual acuity distribution and the amount of vision change between the preoperative, 1-year postoperative, and 3-year postoperative periods. An increase of ≥0.3 logMAR unit, a change of <0.3 logMAR unit, and a decrease of ≥0.3 logMAR unit in comparison with the preoperative value were defined as “worsenig,” “invariant,” and “improvement,” respectively.

7) In the discussion, if the authors elect to compare success rates between studies, they should make it clear in the text and the table that every study used in the comparison relied on the same success criteria as the present study. If this is not the case, the authors may still choose to report the success rates of these studies, but should discuss the impact of definition discrepencies. They may also consider reporting more comparable factors such as the percentages of intraocular pressure and medication reduction.

Response: We thank the reviewer for this comment. The definition of surgical success in each study is given in footnotes.

Page 17, Lines 238-239

a Surgical success was defined as 22 mmHg> IOP >5 mmHg. 

b surgical success was defined as 22 mmHg> IOP.

Moreover, the issues of comparative research by surgical technique are described in Limitation.

Page 19, Lines 283-289

However, the present study has several limitations, including its retrospective design and lack of a control group. An additional period of follow-up is required to assess long-term prognosis, including tube exposure. Comparison with TLE, the types of glaucoma implants, glaucoma surgery history, and combination with anti-VEGF treatment should be further evaluated. As administration of anti-VEGF to NVG is currently permitted in Japan, a thorough examination of the administration protocol and investigation of its effectiveness in primary glaucoma implant surgery are necessary.

8) Occasional syntax errors may be addressed through careful proofreading.

Response: We thank the reviewer for reviewing our study. This manuscript has been edited and rewritten by an experienced scientific editor, who has improved the grammar and stylistic expression of the paper.

 

Reviewer #3: This is was a retrospective, single-center, single surgeon pair case series aiming to describe outcomes in patients undergoing primary non-valved pars plana tube placement with concomitant vitrectomy in patients with neovascular glaucoma. This study is of some clinical interest as these surgeries are not frequently done at the same time and valved glaucoma tube shunts are perhaps more frequently used in patients with NVG.

Response: We wish to thank the reviewer for reviewing our study. We have revised manuscript according to REVIEWER 3.

Abstract: With regards to abstract, it would be helpful to define the success/failure criteria as it is only described later. I think too much space in the abstract is given to a breakdown of the complications, which could better be discussed later.

Response: We thank the reviewer for this comment. Accordingly, we have added the definition of success in Abstract as follow:

Page 2, Lines 23-24 (Abstract)

The surgical success of the BGI was defined as an IOP of <22 mmHg and <5 mmHg with or without antiglaucoma medication.

Introduction: The paper as a whole would benefit from proof-reading by a native English speaker. However, this does not detract from the underlying readability, except in a few instances (particularly at lines 104-106).

Response: We thank the reviewer for reviewing our study. This manuscript has been edited and rewritten by an experienced scientific editor, who has improved the grammar and stylistic expression of the paper. Also we have revised the pointed sentence as follow:

Page 7, Line 101-103

Moreover, informations on intraoperative complications, postoperative complications, reoperation, and reasons for treatment failure were collected.

For my own clarification, lines 45-48 seem to suggest that the TVT study showed the same outcomes, when I would argue they were similar and there were differences between the tube and trabeculectomy groups.

Response: We thank the reviewer for this comment. The reviewer's comment is correct. We have revised the manuscript in Introduction as follows:

Page 4, Line 46-49 (Introduction)

The choice of surgical treatment for glaucoma has expanded in recent years, and tube shunt surgery, which is one of the treatments for intractable glaucoma, has the similar effect of lowering IOP as trabeculectomy in the Tube versus Trabeculectomy (TVT) Study [2-4].

Materials and Methods: It might be more statistically sound to only use 1 eye from any 1 patient. In this study there are several patients that have both eyes enrolled.

Response: We thank the reviewer for this comment. We received a similar point from Reviewer 3. Accordingly, we have exclude 5 eyes and revised the manuscript including Table 1-4 and Figure 2-3. The results do not change much.

It would of course be up to the editor, but the lengthy description of the surgical technique may be better suited for an addendum as it does not add much to the paper.

Response: We thank the reviewer for this comment. There are various variations such as the thread used for BGI insertion and the presence or absence of a scleral patch, and we considered that it was important to describe the details of the surgical procedure. We would like to leave this point to the editor's judgment.

Discussion: The success rates of the Baerveldt, Ahmed, and trabeculectomy are pulled from a few limited case series and I am concerned that there is the risk for bias. Success may be measured differently between these studies and the authors run the risk of subconsciously choosing studies that support their conclusion that the Baerveldt is highly successful.

Response: We thank the reviewer for this comment. The definition of surgical success in each study is given in footnotes.

Page 17, Lines 238-239

a Surgical success was defined as 22 mmHg> IOP >5 mmHg. 

b surgical success was defined as 22 mmHg> IOP.

Moreover, the issues of comparative research by surgical technique are described in Limitation.

Page 19, Lines 283-289

However, the present study has several limitations, including its retrospective design and lack of a control group. An additional period of follow-up is required to assess long-term prognosis, including tube exposure. Comparison with TLE, the types of glaucoma implants, glaucoma surgery history, and combination with anti-VEGF treatment should be further evaluated. As administration of anti-VEGF to NVG is currently permitted in Japan, a thorough examination of the administration protocol and investigation of its effectiveness in primary glaucoma implant surgery are necessary.

While the authors acknowledge their reasons for not using anti-VEGF pre/post-operatively, the standard of care for NVG now almost always includes anti-VEGF. The conclusions of the study may be outdated because of this.

Response: We thank the reviewer for this comment. We have discussed the use of anti-VEGF in NVG in discussions as follows:

Page 19, Lines 262-269 (Discussions)

The effectiveness of the adjuvant use of anti-VEGF has been reported in TLE for NVG [15, 16]. Zhou et al. reported that anti-VEGF is effective for the treatment success and prevention of bleeding complications in a meta-analysis [19]. On the other hand, Noor et al. reported no significant differences in the postoperative IOP, number of glaucoma medications, and success rate with or without the use of anti-VEGF in tube shunt surgery for NVG [17]. Anti-VEGF was not used in this study because it had not been approved during the study period. On the basis of previous reports, the use of anti-VEGF may reduce bleeding complications in BGI surgery, but further studies are needed on the final therapeutic effect.

Based on current study, which showed the therapeutic results of Primary tube without anti-VEGF, we can expect the development of research on the therapeutic results of anti-VEGF combination in the future. We have discussed this point as follows:

Page 19, Lines 285-289 (Discussions)

Comparison with TLE, the types of glaucoma implants, glaucoma surgery history, and combination with anti-VEGF treatment should be further evaluated. As administration of anti-VEGF to NVG is currently permitted in Japan, a thorough examination of the administration protocol and investigation of its effectiveness in primary glaucoma implant surgery are necessary.

The study may benefit from more comparison of complication rates with other tube insertion techniques (like anterior chamber placement)

Response: We thank the reviewer for this comment. We have discussed the comparison of complication rates with other studies in discussions as follows: 

Page 17, Line 249 to Page 18, Line 261

The TVT study [2] reported the following complication rates: hyphema, 2%; vitreous hemorrhage, 1%; endophthalmitis, 1%; tube erosion, 5%; and retinal detachment, 1%. Meanwhile, the PTVT study [10] reported 6% and 1% incidence rates of hyphema and plate exposure, respectively. In addition, Anton et al [18] reported the following incidence rates: hyphema, 21%; vitreous hemorrhage, 16%; endophthalmitis, 3.4%; and retinal detachment, 4.5%, in BGI surgery with vitrectomy. In tube shunt surgery for NVG, the incidence rates of vitreous hemorrhage and hyphema seem high. In this study, the incidence rates of vitreous hemorrhage and hyphema were 18.8% and 18.8%, respectively (Table 3). On the other hand, no complications of endophthalmitis, retinal detachment, and plate/tube exposure were found in this study (Table 3). We believe that sharing treatment strategies in the fields of glaucoma and vitreoretina from the initial surgery is the factor of success. The absence of plate/tube exposure as a complication may have resulted from the adequate surgical field and use of preserved sclera for patching in all the cases.

Lines 253-254 are difficult to understand

Response: We thank the reviewer for this comment. We have revised the pointed sentence as follow:

Page 17, Line 243-244

Primary tube surgery is easier to insert glaucoma implants than secondary tube surgery after TLE. The reason is that the conjunctiva is preserved in primary tube surgery. This point is considered to be an advantage of primary tube surgery.

Lines 254-256: The reasoning for the lack of tube exposure after only 3 years may be of limited utility as tube exposure often happens even further out than 3 years

Response: We thank the reviewer for this comment. The reviewer's comment is correct. We have added the need for long-term observation in Discussions as follows:

Page 19, Lines 284-285 (Discussions)

An additional period of follow-up is required to assess long-term prognosis, including tube exposure.

Only one small study is cited for the superiority of pars plana tubes in protecting corneal endothelium.

Response: We thank the reviewer for this comment. We have added literature showing that the anterior chamber insertion type is disadvantageous for the reduction of the corneal endothelium as follows:

Page 19, Lines 277-279

Zhang et al. compared more than 100 Ahmed valve implant cases of sulcus and anterior chamber tube placement and showed that tube location in anterior chamber were associated with faster central endothelial cell density loss[22]. 

Overall, the conclusions drawn by the authors in this study suggest that combined non-valved pars plana tube placement in conjunction with vitrectomy is successful at lowering IOP with relatively low complication rates.

Response: We thank the reviewer for this comment. We have revised the conclusions in Abstract and Discussions as follows:

Page 3, Line 39-40 (Abstract)

In conclusion, combined non-valved pars plana tube placement in conjunction with vitrectomy was successful at lowering IOP with relatively low complication rates.

Page 19, Line 290 to page 20 Line 294 (Discussions)

In summary, we evaluate the 3-year long-term outcomes of primary BGI surgery for NVG. Our results suggest that combined non-valved pars plana tube placement in conjunction with vitrectomy is successful at lowering IOP with relatively low complication rates. In the strategy of primary tube surgery combined with vitrectomy, collaboration between glaucoma and retina specialists may be important.

---

## [Editor Report · Decision Letter 1]

29 Mar 2021

Surgical outcomes after primary Baerveldt glaucoma implant surgery with vitrectomy for neovascular glaucoma

PONE-D-21-04967R1

Dear Dr. Nishitsuka,

We’re pleased to inform you that your manuscript has been judged scientifically suitable for publication and will be formally accepted for publication once it meets all outstanding technical requirements.

Kind regards,

Ahmed Awadein, MD, Ph.D, FRCS

Academic Editor

PLOS ONE
---

## [Editor Report · Acceptance letter]

5 Apr 2021

PONE-D-21-04967R1 

Surgical outcomes after primary Baerveldt glaucoma implant surgery with vitrectomy for neovascular glaucoma 

Dear Dr. Nishitsuka:

I'm pleased to inform you that your manuscript has been deemed suitable for publication in PLOS ONE. Congratulations! Your manuscript is now with our production department. 

Kind regards, 

on behalf of

Dr. Ahmed Awadein 

Academic Editor

PLOS ONE